# Prognostic Significance of *TWIST1*, *CD24*, *CD44*, and *ALDH1* Transcript Quantification in EpCAM-Positive Circulating Tumor Cells from Early Stage Breast Cancer Patients

**DOI:** 10.3390/cells8070652

**Published:** 2019-06-29

**Authors:** Areti Strati, Michail Nikolaou, Vassilis Georgoulias, Evi S. Lianidou

**Affiliations:** 1Analysis of Circulating Tumor Cells Lab, Department of Chemistry, University of Athens, 15771 Athens, Greece; 2Medical Oncology Unit, “Elena Venizelou” Hospital, 11521 Athens, Greece; 3Metropolitan General Hospital, 15562 Athens, Greece

**Keywords:** liquid biopsy, circulating tumor cells, epithelial–mesenchymal transition, stem cells, early breast cancer

## Abstract

(1) Background: The aim of the study was to evaluate the prognostic significance of EMT-associated (*TWIST1)* and stem-cell (SC) transcript (*CD24*, *CD44*, *ALDH1*) quantification in EpCAM+ circulating tumor cells (CTCs) of early breast cancer patients. (2) Methods: 100 early stage breast cancer patients and 19 healthy donors were enrolled in the study. *CD24*, *CD44*, and *ALDH1* transcripts of EpCAM^+^ cells were quantified using a novel highly sensitive and specific quadraplex RT-qPCR, while *TWIST1* transcripts were quantified by single RT-qPCR. All patients were followed up for more than 5 years. (3) Results: A significant positive correlation between overexpression of *TWIST1* and *CD24^−/low^/CD44^high^* profile was found. Kaplan–Meier analysis revealed that the ER/PR-negative (HR-) patients and those patients with more than 3 positive lymph nodes that overexpressed *TWIST1* in EpCAM^+^ cells had a significant lower DFI (log rank test; *p* < 0.001, *p* < 0.001) and OS (log rank test; *p* = 0.006, *p* < 0.001). Univariate and multivariate analysis also revealed the prognostic value of *TWIST1* overexpression and *CD24^−/low^/CD44^high^* and *CD24^−/low^/ALDH1^high^* profile for both DFI and OS. (4) Conclusions: Detection of *TWIST1* overexpression and stem-cell (*CD24, CD44, ALDH1*) transcripts in EpCAM^+^ CTCs provides prognostic information in early stage breast cancer patients.

## 1. Introduction

Circulating tumor cells (CTCs) are major players in liquid biopsy [1,2], and their molecular characterization is highly important for rational treatment decisions and for monitoring therapeutic response [3], whereas their analysis at the single cell level has the potential to reveal tumor heterogeneity in real time [4]. In breast cancer, a subpopulation of tumor cells that display stem cell-like properties [5] determines the aggressive characteristics and drug resistance of tumor clonal evolution [6]. Cancer stem cells (CSCs) that mediate tumor metastasis and therapeutic resistance have the capacity to transition between mesenchymal and epithelial-like states [7]. It has already been shown that breast cancer cells with the CD44+CD24−/low phenotype [8] that overexpress aldehyde dehydrogenase 1 (ALDH1+) [9] are able to form tumors in mice with high tumorigenic capacity. It has also been shown that disseminated tumor cells (DTCs) [10] and CTCs express the putative stem cell CD44+/CD24− and/or ALDH1+/CD24− phenotypical profile [11,12]. Moreover, in primary human luminal breast cancer, the metastasis-initiating cells containing CTC that express EPCAM, CD44, CD47, and the proto-oncogene MET are related with reduced overall survival (OS) [13]. In other types of cancer, various stem cell markers have also been identified and correlated with metastatic capacity [14] and poor prognosis [15].

It is now known that breast cancer stem cells exist in distinct mesenchymal-like (epithelial–mesenchymal transition [EMT]) as CD44+/CD24− and epithelial-like (mesenchymal-epithelial transition [MET]) states that express ALDH1. This transition between EMT- and MET-like states is highly important for their capacity to invade, disseminate, and grow at metastatic sites [16]. Many studies have already shown that a major proportion of CTC express both EMT and tumor stem cell characteristics [17,18,19]. Recently it was shown that an EpCAM-/ALDH1+/HER2+/EGFR+/HPSE+/Notch1+ profile in CTC drives these cells to metastasize to the brain [20]. At the single cell level, it has been shown that CTC that co-express the stem cell marker ALDH1 and the mesenchymal marker *TWIST1* may prevail during disease progression [21]. However, the prognostic significance of EMT and Stem cell (SC) markers in CTC has only been shown up to now in metastatic colorectal cancer [22] and metastatic breast cancer [23].

In early breast cancer, the molecular detection of cytokeratin 19 (CK-19) mRNA-positive cells in peripheral blood before [24], during [25], and after adjuvant therapy [26] is associated with worse prognosis, while their elimination seems to be an efficacy indicator of treatment [27]. The prognostic significance of CTC count using the CellSearch system in neoadjuvant [28] and adjuvant early breast cancer patients [29] has been also shown. Moreover, the administration of “secondary” adjuvant trastuzumab in patients with HER2(−) breast cancer can eliminate chemotherapy-resistant CK19 mRNA-positive CTCs [30], in contrast to the Treat CTC phase II trial that failed to prove the efficacy of trastuzumab in the detection rate of CTC [31]. However, in early breast cancer stages the early detection of recurrence remains a big challenge [32], and until now, there are not solid data proving the prognostic significance of EMT/SC(+) cells. The aim of the current study was to evaluate the prognostic significance of *TWIST1*, *CD24*, *CD44*, and *ALDH1* mRNA quantification in EpCAM-positive circulating tumor cells from early stage breast cancer patients with a long follow-up.

## 2. Materials and Methods

### 2.1. Cell Lines

The human mammary carcinoma cell line SKBR-3 was used as a positive control for the development of the quadraplex RT-qPCR assay for *CD24*, *CD44*, *ALDH1*, *HPRT*, while MDA-MB-231 cancer cell line was used as a positive control for the expression of *TWIST1* [33]. Cells were counted in a hemocytometer and their viability was assessed by trypan blue dye exclusion. cDNAs of all cancer cell lines were kept in aliquots at −20 °C and used for the analytical validation of the assay, prior to the analysis of patient’s samples.

### 2.2. Patients

In total, 100 patients with non-metastatic breast cancer from the Medical Oncology Unit “Elena Venizelou” Hospital and IASO General hospital were enrolled in the study from September 2007 until January 2013. Peripheral blood (20 mL) was obtained from all these patients two weeks after the removal of the primary tumor and before the initiation of adjuvant chemotherapy. The chemotherapeutic adjuvant treatment for these patients has been previously reported [34]. The clinical characteristics for these patients at the time of diagnosis are shown in Appendix A. All patients signed an informed consent to participate in the study, which was approved by the Ethics and Scientific Committees of our Institutions. Peripheral blood (20 mL) was obtained from 19 healthy female blood donors (HD) and was analyzed in the same way as patients’ samples (control group).

### 2.3. Isolation of EpCAM+ CTCs

To reduce blood contamination by epithelial cells from the skin, the first 5 mL of blood were discarded, and the blood collection tube was at the end disconnected before withdrawing the needle. Peripheral blood (20 mL in EDTA) from (HD) and patients was collected and processed within 3 h in exactly the same manner. After collection, peripheral blood was diluted with 20 mL phosphate buffered saline (PBS, pH 7.3), and peripheral blood mononuclear cells (PBMCs) were isolated by gradient density centrifugation using Ficol-Paque TM PLUS (GE Healthcare, Bio-Sciences AB) at 670 g for 30 min at room temperature. The interface cells were removed and washed twice with 40 mL of sterile PBS (pH 7.3, 4 °C), at 530 g for 10 min. EpCAM+ cells were enriched using immunomagnetic Ber-EP4 coated capture beads (Dynabeads^®^ Epithelial Enrich, Invitrogen, Carlsbad, CA, USA), according to the manufacturer’s instructions [33].

### 2.4. RNA Extraction-cDNA Synthesis

Total RNA isolation was performed using TRIZOL-LS (ThermoFischer, Carlsbad, CA, USA). All RNA preparation and handling steps took place in a laminar flow hood under RNAse-free conditions. The isolated RNA from each fraction was dissolved in 20 μL of RNA storage buffer (Ambion, ThermoFischer, USA) and stored at −70 °C until use. RNA concentration was determined by absorbance readings at 260 nm using the Nanodrop-1000 spectrophotometer (NanoDrop, Technologies, Wilmington, DE, USA). mRNA was isolated from the total RNA using the Dynabeads mRNA Purification kit (ThermoFischer, USA), according to the manufacturer’s instructions. cDNA synthesis was performed using the High capacity RNA-to-cDNA kit (ThermoFischer, USA) in a total volume of 20 μL, according to the manufacturer’s instructions.

### 2.5. RT-qPCR

A novel quadraplex RT-qPCR assay was first developed for *CD24*, *CD44*, *ALDH1*, and *HPRT* (reference gene). Primers and dual hybridization probes were de novo in-silico designed, using Primer Premier 5.0 software (Premier Biosoft, Palo Alto, CA, USA). The specificity of all primer and hybridization probe sequences was first tested by homology searches in the nucleotide database (NCBI, nucleotide BLAST). Cross reaction between all oligonucleotide sequences was also examined. Each probe set included a 3′-fluorescein (F) donor probe and a 5′-LC acceptor probe that was different for each gene set: *CD24* (610 nm), *CD44* (640 nm), *ALDH1* (670 nm) and *HPRT* (705 nm). A color compensation test was performed by using pure dye spectra so that spectral overlap between dyes was corrected [35]. Quadraplex RT-qPCR reactions were performed in the LightCycler 2.0 (Roche, Mannheim, Germany). Component concentrations and the cycling conditions for the quadraplex RT-qPCR assay were optimized in detail. The amplification reaction mixture (10 μL) contained 1 μL of the PCR Synthesis Buffer (5Χ), 2.4 μL of MgCl_2_ (25 mM), 0.2 μL dNTPs (10 mM), 0.8 μL BSA (10 μg/μL), 0.1 μL Hot Start DNA polymerase (HotStart, 5 U/μL, Promega, Madison, WI, USA), 0.5 μL of a mixture containing all eight primers (10 μΜ), 0.5 μL of a mixture containing all eight dual hybridization probes (3 μM), and H2O (added to the final volume). Cycling conditions of the *CD24*, *CD44*, *ALDH1*, *HPRT* quadraplex RT-qPCR assay were: 95 °C/2 min; 45 cycles of 95 °C/20 s, annealing at 59 °C/20 s, and extension at 72 °C/20 s. For the development and analytical validation of the novel quadraplex RT-qPCR assay, we generated individual PCR amplicons corresponding to the gene-targets studied that served as quantification calibrators, as we have previously described [33]. RT-qPCR for *TWIST1* was performed as previously described [33,36]. All data were evaluated in respect to *TWIST1*, *CD24*, *CD44*, and *ALDH1* expression by normalizing the EpCAM+ fraction of PBMCs to the expression of *HPRT* and the 2–ΔΔCt approach, as described in detail by Livak and Schmittgen [37]. A cut-off value was calculated as the mean of signals derived by samples of healthy individual analyzed in exactly the same way plus 2SD for *TWIST1*, *CD44*, and *ALDH1* transcripts and as the mean of signals derived by samples of healthy individual minus 2SD for *CD24*.

### 2.6. Statistical Analysis

Statistical analysis was performed using SPSS (SPSS Statistics 25.0, company, Armonk, NY, USA). The chi-square test of independence or Fisher exact test (SPSS, version 25.0) was used to make comparisons between groups. The DFI and OS rate were calculated by the Kaplan–Meier method and were evaluated by the log-rank test. Cox proportional hazards (PH) models were used to evaluate the relationship between EMT and Stem Cell status and event-time distributions, with tumor size, grade, number of involved lymph nodes, ER, PR, HER2, and age. Parametric and non-parametric tests were used to compare continuous variables between groups. All P-values are two-sided. A level of *p* < 0.05 is considered statistically significant.

## 3. Results

### 3.1. Analytical Validation of the Quadraplex RT-qPCR Assay for CD24, CD44, ALDH1, HPRT

The analytical specificity of the developed assay was checked by using all oligonucleotides in a common master mix in four different reactions in the presence of one individual gene target each time. Each primer pair and dual hybridization probe pair amplifies specifically only the corresponding target sequence and is detected only in the corresponding wavelength (Appendix A). The analytical sensitivity was determined for each individual gene target using a calibration curve. These calibration curves were generated using serial dilutions of individual gene-specific external standards in triplicate for each concentration, ranging from 10^5^ copies/μL to 10 copies/μL. The analytical detection limit corresponded to 3 copies/μL while the quantification limit was equal to 9 copies/μL (Appendix A). The developed assay showed linearity over the entire quantification range and correlation coefficients greater than 0.99 in all cases, indicating a precise log-linear relationship. Intra and inter-assay variance: Repeatability or intra–assay variance of the quadraplex RT-qPCR was evaluated by repeatedly analyzing four cDNA samples corresponding to 1, 10, 100, and 1000 SKBR-3 cells in the same assay, in three parallel determinations. Reproducibility or interassay variance was evaluated by analyzing the same cDNA sample, representing 1000 SKBR-3 cells on five separate assays performed in five different days (Appendix A).

### 3.2. Quantification of CD24, CD44, ALDH1, and TWIST1 mRNA in the EpCAM(+) Fraction in Early Stage BrCa Patients and (HD) 

In all EpCAM(+) fractions isolated from 100 early BrCa patient samples and 19 HD *CD24*, *CD44*, *ALDH1*, *HPRT* transcripts were quantified by the developed quadraplex RT-qPCR and *TWIST1* transcripts by the singleplex RT-qPCR assay (Figure 1). Median fold change of *TWIST1* expression in the EpCAM(+) fraction was 0.42 (range: 0–0.95) in HD and 10.06 (range: 2.33–3327) in *TWIST1^high^* (Mann-Whitney test, Ζ = −1.363, *p* = 0.001) and 0 (range: 0–0) in TWIST1^low/−^ early BrCa patient samples (Mann-Whitney test, Ζ = −3.634, *p* < 0.001) (Figure 1A). Median fold change of *CD24* expression in the EpCAM(+) fraction was 2.00 (range: 1.42–3.81) in HD and 1.91 (range: 0.91–15.14) in *CD24*^high^ (Mann-Whitney test, Ζ = −0.492, *p* = 0.623) and 0.62 (range: 0.29–0.88) in *CD24*^low^ early BrCa patients (Mann-Whitney test, Ζ = −5.577, *p* < 0.001) (Figure 1B). Median fold change of *CD44* expression in the EpCAM(+) fraction was 0.71 (range: 0.14–1.06) in HD and 2.33 (range: 1.28–202.75) in *CD44^high^* (Mann-Whitney test, Ζ = −6.084, *p* < 0.001) and 0.61 (range: 0.01–1.17) in *CD44^low^* early BrCa patients (Mann-Whitney test, Ζ = −1.084, *p* = 0.278) (Figure 1C). Median fold change of *ALDH1* expression in the EpCAM(+) fraction was 1.32 (range: 0.69–2.19) in HD and 2.97(range: 2.30–14.72) in *ALDH1^high^* (Mann-Whitney test, Ζ = −5.119, *p* < 0.001) and 0.84 (range: 0.06–2.16) in *ALDH1^low^* early BrCa patients (Mann-Whitney test, Ζ = −2.190, *p* = 0.029) (Figure 1D).

In 19/100(19%) breast cancer samples tested, *TWIST1* was overexpressed, while in 15/100(15%) samples the *CD24^−/low^/CD44^high^* profile, and in 9/100(9%) the *CD24^−/low^/ALDH1^high^* profile was detected (Figure 2A). There was a positive correlation between *TWIST1* mRNA overexpression and the *CD24^−/low^/CD44^high^* profile (Fisher’s Exact Test; *p* = 0.008), while there was no correlation between *TWIST1* mRNA overexpression and the *CD24^−/low^/ALDH1^high^* profile (Fisher’s Exact Test; *p* = 0.366) (Table 1). *TWIST1* overexpression and *CD24^−/low^/CD44^high^* and/or *CD24^−/low^/ALDH1^high^* were detected in 7/100(7%) EpCAM(+) samples. The correlation between these characteristics and the clinical variables of the patients revealed an association between *TWIST1* overexpression with lymph node status (chi-square; *p* = 0.036) and HER2 status of the primary tumor (chi-square; *p* = 0.006) (Appendix A).

### 3.3. Evaluation of Prognostic Significance

#### 3.3.1. Disease Free Interval

During the follow up period (median: 95 months; range: 4–137 months), 25/100 (25%) patients relapsed and in 9/25 (36%) of them *TWIST1* overexpression was detected in the EpCAM+ CTC fraction (Fisher’s Exact Test; *p* = 0.019). Similarly, 6/25 (24%) patients displayed a Stem Cell profile in EpCAM+ CTC fraction (Fisher’s Exact Test; *p* = 0.194). In 4/25 (16%) of these patients, both *TWIST1* overexpression and the Stem Cell profile was detected (Fisher’s Exact Test; *p* = 0.063) (Appendix A). The Kaplan–Meier estimates of the cumulative DFI of the patients overexpressing *TWIST1* revealed that these patients had worse survival compared to patients who were negative (83.6mo vs 115.8mo respectively; *p* = 0.019) (Table 2, Figure 3A). However, the stem cell profile alone (86.7mo. vs 113.2mo, respectively in the two groups; log rank test; *p* = 0.174) (Table 2, Appendix A) and both stem cell and mesenchymal characteristics (68.9mo vs 88.8–115.8mo, respectively; *p* = 0.087) (Table 2, Appendix A) failed to show any statistically significant difference even though the mean survival showed a reduced trend. Kaplan–Meier survival analysis of patients with positive axillary lymph nodes and *TWIST1* mRNA overexpression had worst DFI (Table 2, Appendix A) (82.6 mo. vs 88.7–123.3; *p* = 0.05). When all patients were divided into two groups based on the number of positive lymph nodes (1–3, and ≥4 positive nodes) and the overexpression of *TWIST1* [(N_2-3_/*TWIST1*(+), N_2-3_/*TWIST1*(−), N_1_/*TWIST1*(+), and N_1_/*TWIST1*(−)], Kaplan–Meier analysis revealed that women harboring more than 3 positive lymph nodes and *TWIST1* that was overexpressed in EpCAM+ CTC fraction had a statistically significant shorter DFI (Table 2, Figure 3C) (mean survival: 68.6mo vs. 103.0–114.3mo.; *p* = 0.007). When patients were dichotomized accordingly to the HR status (ER/PR) in the following groups: a) HR(−)/*TWIST1*(+), b) HR(−)/ *TWIST1*(−), c) HR(+)/*TWIST1*(+) and d) HR(+)/*TWIST1*(−), it was observed that women with HR(−)/*TWIST1*(+) profile were characterized by statistically significant shorter DFI (36mo. vs 102.3–117.9mo.; *p* < 0.001; Figure 3E). A Univariate analysis (Table 3) also revealed the significance of (a) *TWIST1*(+), (b) HR(−)/*TWIST1*(+), (c) *TWIST1*(+) /N_2-3_, d) SC (+)/ *TWIST1*(+) (Figure 2B) in the risk of disease progression. Multivariate confirmed the prognostic value of HR(−)/*TWIST1*(+) and *TWIST1*(+)/N2-3, in the EpCAM(+) CTC fraction for the prediction of DFI (Table 3) independently from patients’ age, tumor T stage, grade, nodal status alone and the HR, and HER2 status of the primary tumor.

#### 3.3.2. Overall Survival

Among the 25 patients that relapsed during the follow up period, 14/25 (56.0%) patients died and 11/25 (44.0%) were still alive at the time of the last follow-up. In 6/14 (42.9%) patients that died *TWIST1* overexpression was detected in the EpCAM+ fraction (Fisher’s Exact Test; *p* = 0.024). Similarly, 4/14 (28.6%) patients displayed a Stem Cell profile in EpCAM+ CTC fraction (Fisher’s Exact Test; *p* = 0.217). In 3/14 (21.4%) of these patients, both *TWIST1* overexpression and *CD24^−/low^/CD44^high^* and/or *CD24^−/low^/ALDH1^high^* profiles (Fisher’s Exact Test; *p* = 0.055) were detected (Fisher’s Exact Test; *p* = 0.055) (Appendix A). The Kaplan–Meier estimates of the overall survival (OS) of the patients overexpressing *TWIST1* were significantly different in favor of patients who were negative for *TWIST1* overexpression (106.4 vs 127.2 mo; *p* = 0.046) (Table 2, Figure 3B). Stem Cell profiles (107.3 vs 125.2 mo.; *p* = 0.171) (Table 2, Appendix A) and the co-expression of EMT and SC-associated genes (96.29 vs 109.1–127.3 mo.; *p* = 0.118) (Table 2, Appendix A) failed to show any statistically significant difference. There was no difference in OS in patients with *TWIST1* overexpression according to N0 and N+ lymph node involvement (108.8 mo vs 92–129 mo, respectively; *p* = 0.194; Appendix A). However, when the Kaplan–Meier curves for OS for *TWIST1* overexpression were additionally stratified according to lymph nodes status (Table 2, Figure 3D) and HR status (Table 2, Figure 3F) our data have shown that patients with >3 LN and *TWIST1* overexpression had lower OS (109.8 mo., range: 115–129 mo.; *p* = 0.026); the same was seen for patients that were HR(−) and *TWIST1* was overexpressed (65.7 vs 110.2–131.9 mo.; *p* < 0.001). Univariate analysis showed a significantly higher risk of death in the group of patients positive for *TWIST1* overexpression that had more than 3 lymph nodes affected or co-expressed the stem cell profile (Figure 2B). Multivariate analysis confirmed the prognostic value of *TWIST1* overexpression in combination with N_2-3_, and in combination with HR(−) status in the EpCAM(+) CTC fraction for the prediction of OS, independently from patients’ age, tumor T stage, grade, nodal status, and the status of the receptors ER, PR, HER2 of the primary site (Table 3).

## 4. Discussion

Molecular characterization of CTCs at the gene expression level has a strong potential to provide novel prognostic and predictive biomarkers. It is now clear through numerous studies that CTCs isolated from breast cancer patients express epithelial markers [38], receptors (ER, PR, HER2, EGFR), stem cell markers [39], and mesenchymal markers [11]. So far, most studies have been performed in the metastatic setting where the number of circulating tumor cells is usually high. However, in the non-metastatic setting of breast cancer, CTCs are not always detected and their numbers are usually very low, thus their molecular characterization is extremely difficult. For this reason, in the early breast cancer setting, a higher volume of peripheral blood used for the analysis of CTCs is very critical. Our group has shown many years ago the prognostic significance of *CK-19* mRNA detection in peripheral blood of early breast cancer patients, using 20 mL of peripheral blood for CTC isolation and further downstream analysis [36,38]. Other groups have also shown that the detection of CTCs in the early breast cancer setting is providing critical prognostic information for these patients [29].

In this study we evaluated for the first time the prognostic significance of *TWIST1, CD24, CD44,* and *ALDH1* transcript quantification in EpCAM-positive circulating tumor cells isolated from peripheral blood of early stage breast cancer patients. We selected *TWIST1* as this is a very established EMT marker; for this reason, we have developed already in 2011 an RT-qPCR assay for the absolute quantification of *TWIST1*-mRNA expression, and we have validated this assay in EpCAM-positive cells isolated by early and metastatic breast cancer patients [33]. Concerning the selection of stem cell markers, this was based on publications by Al-Hajj M et.al. [8] and Ginestier C et.al. [9], who have shown that the breast cancer stem cell phenotypes of (a) CD44^+^*/CD24*^−/low^ phenotype and (b) the overexpression of aldehyde dehydrogenase 1 (ALDH1+) are able to form tumors in mice with high tumorigenic capacity.

Multiplex RT-qPCR assays have many benefits due to their wide dynamic range, the low sample volume required, and the reduced time of analysis [35]. Our study was based on an analytically validated novel multiplex assay for the quantification of *CD24*, *CD44*, *ALDH1,* and *HPRT* and a single RT-qPCR assay for the quantification of *TWIST1* transcripts. The analytical sensitivity and specificity of the novel quadraplexRT-qPCR assay for the simultaneous detection of *CD24*, *CD44*, *ALDH-1,* and *HPRT* transcripts were determined by using calibrators specific for each gene target. Both these assays were validated according to the Minimum Information for Publication of Quantitative Real-Time PCR Experiments (MIQE) guidelines [40].

Relevant prognostic and predictive markers in early breast cancer cohort is of major significance. The SUCCESS A trial has shown that the presence of CTCs, as evaluated in 30 mL of peripheral blood, two years after chemotherapy has been associated with decreased OS and DFS in high-risk early breast cancer patients [41]. Lucci et.al. has also shown that the presence of one or more circulating tumor cells could predict early recurrence and decreased overall survival in chemonaive patients with non-metastatic breast cancer [29]; however, the main limitation of this study is that it was based on CTC enumeration performed in only 7.5 mL of blood. Additionally, molecular characterization of CTC could identify CTC biomarkers that are associated to specific signaling pathways like EMT or CSC. Our findings demonstrate a positive correlation between *TWIST1* overexpression and the *CD24^−/low^/CD44^high^* profile in the EpCAM positive CTC fraction. This is in agreement with previous findings showing that the mesenchymal-like breast cancer stem cells are characterized as CD24^−^/CD44^+^, while the epithelial-like breast cancer stem cells express high levels of aldehyde dehydrogenase (ALDH) [16]. Univariate analysis revealed a significantly higher risk of relapse and death in the group of patients that expressed both stem cell and mesenchymal characteristics. Mego et al. have shown that patients with *TWIST1*-high tumors had a significantly higher percentage of breast cancer stem cells than patients with *TWIST1*-low tumors [19]. Recently, it was shown that in CTC of NSCLC patients the CD44(+)/CD24(−) population possess epithelial–mesenchymal transition characteristics [42], while another study in metastatic colorectal cancer has shown the prognostic significance of CTC that express both EMT and stem-like genes [22]. At the single CTC-level, Papadaki et.al have shown that CTCs expressing high levels of ALDH1 along with nuclear TWIST expression are more frequently detected in patients with metastatic breast cancer [21] and that these cells represent a chemo-resistant subpopulation with an unfavorable outcome [23]. The main limitation of our study is that we examined the expression of only one EMT marker, *TWIST1* in the EpCAM+ cells of early breast cancer patients. Since there is a high heterogeneity in CTC, it may be possible that we have not detected CTCs that express other mesenchymal markers like Vimentin or Snail. We plan to extend this study by adding more gene expression markers in a biggest sample cohort and correlate our results to the clinical outcome.

According to our results, patients with *TWIST1* overexpression in the EpCAM+ CTCs fraction and more than 3 involved lymph nodes had a significant lower DFI and OS. Similar to our results, recently, Emprou C et al. have shown that in frozen NSCLC tumor samples *TWIST1* is more frequently overexpressed in the N+ group compared to the N0 group showing that partial EMT is involved in lymph node progression in early stages of NSCLC [43], while in primary breast cancer loss of E-cadherin is correlated with more than 3 LN involved in 80% of the patients [44]. Our results also indicate that patients with *TWIST1* overexpression in the EpCAM+ CTCs fraction and a hormone receptor-negative primary tumor had a worse prognosis both for DFI and OS. This is in accordance with previous findings that have shown that the estrogen receptor silencing induces epithelial to mesenchymal transition in breast cancer [45]. It has also been previously shown that in human breast tumors there is an inverse relationship between *TWIST1* and ER expression that may possibly contribute to the generation of hormone-resistant, ER-negative breast cancer [46]. It has also been reported that EMT likely occurs in the basal-like phenotype both in MCF10A cells [47] and in invasive breast cancer carcinomas [48].

## 5. Conclusions

In conclusion, detection of *TWIST1* overexpression and stem-cell (*CD24*, *CD44*, *ALDH1*) transcripts in the EpCAM^+^ CTC fraction provides prognostic information in early stage breast cancer patients. Overexpression of *TWIST1* in the EpCAM^+^ CTC fraction in the group of HR negative patients or with >3 positive lymph nodes is associated with worse prognosis.

## Figures and Tables

**Figure 1 cells-08-00652-f001:**
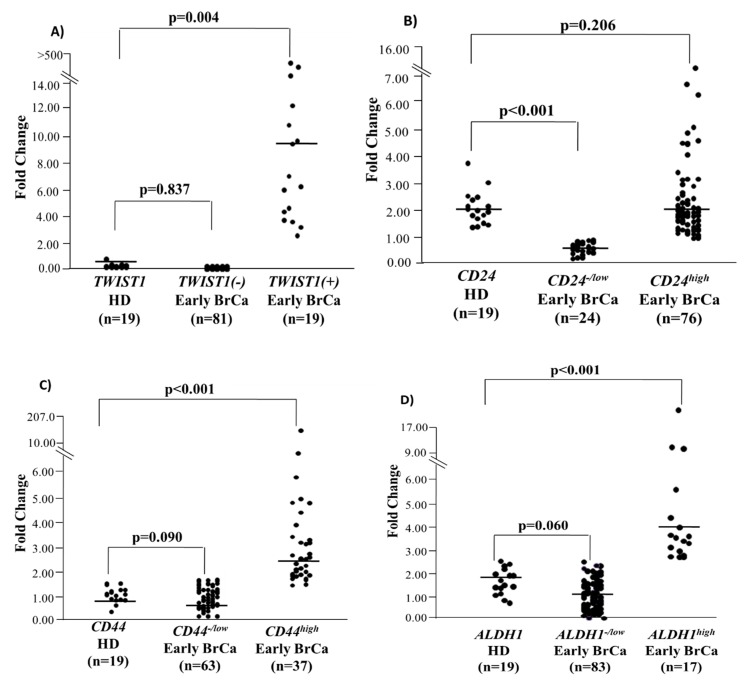
Relative fold change values (2^–ΔΔCt^) in respect to HPRT expression for: (**A**) *TWIST1* (**B**) *CD24*, (**C**) *CD44*, (**D**) *ALDH1* for early breast cancer patients (*n* = 100) and (HD), (*n* = 19).

**Figure 2 cells-08-00652-f002:**
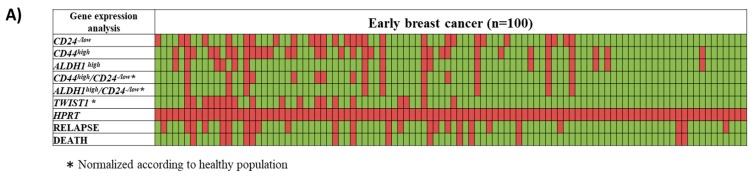
(**A**) Heat map of *TWIST1*, *CD24*, *CD44*, and *ALDH1*-mRNA quantification in the EpCAM+ CTC fraction from early stage breast cancer patients (*n* = 100). Red color represents overexpression, while green color indicates underexpression or lack of expression. Concerning the relapse or death, red color represents the relapse or death, while green color indicates no relapse or alive status. (**B**) Univariate Cox-regression hazard models for TWIST1 overexpression, *CD44^high^/CD24^−/low^*, and *ALDH1^high^/CD24−^/low^* and the co-expression of the mesenchymal profile, *TWIST1*, and the stem cell profile, *CD44^high^/CD24^−/low^*, and *ALDH1^high^/CD24^−/low^*.

**Figure 3 cells-08-00652-f003:**
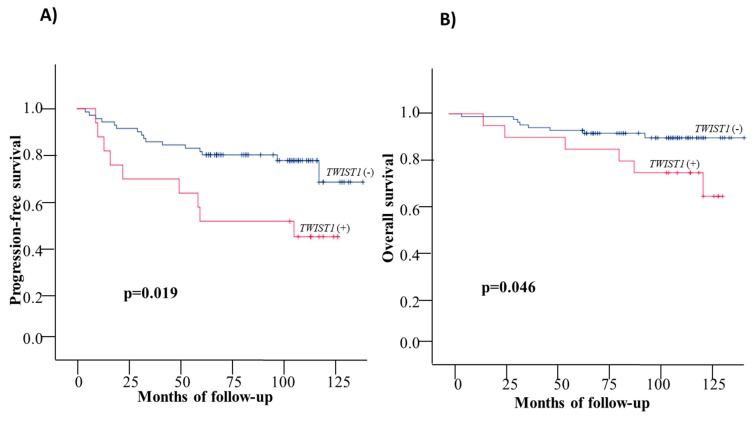
Kaplan–Meier estimates for early BrCa patients: (**A**) DFI: *TWIST1* overexpression, (**B**) OS: *TWIST1* overexpression, (**C**) DFI: *TWIST1* overexpression and number of affected lymph nodes, (**D**) OS: *TWIST1* overexpression and number of affected lymph nodes, (**E**) DFI: *TWIST1* overexpression and HR status, (**F**) OS: *TWIST1* overexpression and HR status.

**Table 1 cells-08-00652-t001:** Correlation between *TWIST1* and *CD44^high^/CD24^−/low^* and *ALDH1^high^/CD24^−/low^* expression in early breast cancer EpCAM positive samples (*n* = 100).

	*CD44^high^/CD24^−/low^*		*p ^a^*	*ALDH1^high^/CD24^−/low^*		*p ^a^*
***TWIST1***	**Positive**	**Negative**	**0.008**	**Positive**	**Negative**	0.366
Positive	7 (46.7%)	12(14.1%)		3 (33.3)	16(17.6%)	
Negative	8 (53.3%)	73(85.9%)		6 (66.7%)	75 (82.4%)	
Concordance	80/100 (80%)			78/100 (78%)		

**^a^** Fischer’s Exact Test. Bold: highlights the significance of the test.

**Table 2 cells-08-00652-t002:** Gene expression in CTCs in respect to DFI and OS.

Gene Expression in CTCs	DFI				OS			
Gene	Mean Survival	95% CI (months)	Range (months)	*p*	Mean Survival	95% CI (Months)	Range (Months)	*p*
*TWIST1+*	83.6	61.9–105.3	9–125	**0.019**	106.4	90.3–122.3	16–127	**0.046**
*TWIST1−*	115	106.3–125.2	4–137		127.2	120.7–133.7	6–137	
Stem cell profile positive (SC+)	86.7	66.7–106.8	16–118	0.174	107.3	89.8–124.8	26–127	0.171
Stem cell profile negative (SC−)	113.2	103.5–122.8	4–137		125.2	118.3–132.1	6–137	
*TWIST1+/SC+*	68.9	39.4–98.31	16–112	0.087	96.2	67.4–125.1	26–127	0.118
*TWIST1+/SC−*	88.8	61.2–116.4	9–125		111.1	93.3–128	16–125	
*TWIST1−/SC+*	100.1	78.6–121.7	41–118		109.1	92.8–125.3	47–118	
*TWIST1−/SC−*	115.8	105.8–126	4–137		127.3	120.4–134.2	6–137	
*TWIST1+/LN+*	82.6	59.3–105.9	9–125	0.05	108.8	93–124.6	16–127	0.194
*TWIST1+/LN−*	88.7	30.5–146.8	16–125		92	39.1–144.8	26–125	
*TWIST1−/LN+*	110.3	99.2–121.4	6–130		121.3	113–129.6	6–130	
*TWIST1−/LN−*	123.3	110.6–135.9	4–137		128.9	120.2–137.7	37–137	
*TWIST1+/N_2–3_*	68.6	40.5–96.7	9–125	**0.007**	98.3	75.5–121.2	16–127	**0.026**
*TWIST1+/N_1_*	104.5	71.8–137.5	13–123		112	94–125	101–118	
*TWIST1−/N_2-3_*	103.1	84.1–121.9	6–130		118.8	103.4–134.1	6–130	
*TWIST1−/N_1_*	114.4	101.7–126.9	9–128		121.1	111.9–130.3	30–128	
*TWIST1+/HR+*	102.3	81.2–123.4	9–125	**<0.001**	121.6	113.9–129.3	79–127	**<0.001**
*TWIST1+/HR−*	36	8.9–63	10–106		65.7	36.7–94.6	76–106	
*TWIST1−/HR+*	117.9	107.1–128.7	4–137		131.9	126.2–137.5	37–137	
*TWIST1−/HR−*	107.7	92.3–123.1	6–127		110.2	96.5–123.84	6–127	

Bold: highlights the significance of the test.

**Table 3 cells-08-00652-t003:** Univariate and multivariate analyses for DFI and OS in the early breast cancer patients group (*n* = 100).

Covariates	Covariate Value	DFI						OS					
		Univariate Cox Regression Analysis			Multivariate Cox Regression Analysis			Univariate Cox Regression Analysis			Multivariate Cox Regression Analysis		
		HR ^a^	95% CI ^b^	*p*	HR ^a^	95% CI ^b^	*p*	HR^a^	95% CI^b^	*p*	HR ^a^	95% CI ^b^	*p*
Age	≥50 vs <50	0.787	0.357–1.734	0.552	0.432	0.169–1.103	0.079	0.718	0.249–2.070	0.539	0.593	0.169–2.080	0.414
*ER*	Yes vs No	0.647	0.286–1.463	0.295	4.391	1.040–18.549	**0.044**	0.238	0.079–0.721	**0.011**	0.623	0.092–4.215	0.628
*PR*	Yes vs No	0.492	0.217–1.114	0.089	0.087	0.021–0.362	**0.001**	0.196	0.054–0.707	**0.013**	0.098	0.011–0.851	**0.035**
*HER2*	Yes vs No	0.500	0.197–1.269	0.145	0.626	0.232–1.693	0.357	0.381	0.106–1.366	0.139	0.247	0.060–1.023	0.054
Lymph nodes	N_0_ vs N_1_ vs N_2–3_	2.207	1.261–3.861	**0.006**	2.433	1.272–4.654	**0.007**	1.659	0.817–3.371	0.162	1.351	0.637–2.862	0.433
Size	≥2cm vs <2cm	3.060	1.049–8.922	**0.041**	4.926	1.225–19.811	**0.025**	7.244	0.947–55.432	0.056	17.450	1.464–208.1	**0.024**
Grade	I/II vs III	1.366	0.570–3.273	0.485	0.753	0.228–2.483	0.641	1.953	0.544–7.008	0.305	0.286	0.040–2.028	0.211
*TWIST1*	Yes vs No	2.582	1.135–5.875	**0.024**	1.382	0.490–3.899	0.540	2.851	0.975–8.33	0.051	1.481	0.382–5.743	0.570
HR and *TWIST1* status	HR+*TWIST1*+	0.486	0.302–0.784	**0.003**	0.597	0.360–0.991	**0.046 ^c^**	0.576	0.313–1.062	0.077	0.666	0.345–1.284	0.225^c^
	HR+*TWIST1*-												
	HR-*TWIST1*+												
	HR-*TWIST1*-												
LN and *TWIST1* status	N_0-1_*TWIST1*+	0.540	0.383–0.762	**<0.001**	0.542	0.371–0.792	**0.002 ^d^**	0.559	0.357–0.875	**0.011**	0.604	0.373–0.976	**0.040 ^d^**
	N_0-1_*TWIST1*-												
	N_2-3_*TWIST1*+												
	N_2-3_*TWIST1*-												
Stem cell profile	Yes vs No	1.873	0.746–4.703	0.181	1.755	0.526–5.855	0.360	2.206	0.690–7.050	0.182	3.689	0.806–16.884	0.093
Stem cell profile/*TWIST1+*	Yes vs No	0.663	0.473–0.929	**0.017**	0.776	0.521–1.146	0.202^e^	0.624	0.402–0.967	**0.035**	0.634	0.378–1.065	0.085^e^

^a^ Hazard ratio, estimated from Cox proportional hazard regression mode. ^b^ Confidence interval of the estimated HR. Results are based on 1000 bootstrap samples and obtained after the Bias corrected and accelerated (BCa) approach. ^c^ Multivariate model adjusted for age, HER2, LN, Size, Grade, Stem cell. ^d^ Multivariate model adjusted for age, ER, PR, HER2, Size, Grade, Stem cell. ^e^ Multivariate model adjusted for age, ER, PR, HER2, LN, Size, Grade. Bold: highlights the significance of the test.

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
