# Peer review of "Prognostic Significance of TWIST1, CD24, CD44, and ALDH1 Transcript Quantification in EpCAM-Positive Circulating Tumor Cells from Early Stage Breast Cancer Patients"

_cells, 2019, doi:10.3390/cells8070652_

Round 1

Reviewer 1 Report

The authors nicely present their study. I did not have any major concerns on this article. I have a minor concern:  

1. The figures are too small making it difficult to read and understand what is being shown.

Author Response

REVIEWER 1

Comments and Suggestions for Authors

The authors nicely present their study. I did not have any major concerns on this article. I have a minor concern: 

1.         The figures are too small making it difficult to read and understand what is being shown.

We would like to thank the reviewer for his positive comments. We have now changed in the revised manuscript all figures, so that they are bigger.

Reviewer 2 Report

The manuscript emphasizes the prognostic relevance of mRNA quantification in EpCAM-positive circulating tumor cells from early stage breast cancer patients.

The authors conclude that detection of TWIST1 overexpression and stem-cell (CD24, CD44, ALDH1) transcripts in the EpCAM+ CTC fraction provides prognostic information in early stage breast cancer patients. Overexpression of TWIST1 in the EpCAM+ CTC fraction in the group of HR negative patients or with >3 positive lymph nodes is associated with worse prognosis.

Comments/Suggestions:

Authors should update the literature on role of CTC in early Breast Cancer and compare the findings in discussion (eg. SUCCESS Study)

Authors should mention about the selection criteria for candidate markers.

How was the cut-off for mRNA expression determined for each marker? Can authors deduce a composite score for independent (not corelated) markers?

Why and how the target genes were dichotomized as “low” and “high” expressors?

Authors may want to comment on limitations and future directions for the study.

What is sensitivity and specificity for extraction of EpCAM+ cells from 20ml of blood? What is RNA yield?

Was control (HPRT) included with TWIST1 gene expression assay in the same reaction? How the mRNA expression was normalized?

Minor Comments:

Authors may provide full form for the abbreviations used first time in text (eg. “HD” to Healthy Donor).

Provide data values consistent across the manuscript (eg- Line 184- provide numbers corresponding to 25%)

Provide detailed figure legend for 2A. Explain the heatmap (what the colors are standing for)?

Provide complete figure legend for Figure 3.

Author Response

REVIEWER 2

Comments and Suggestions for Authors

The manuscript emphasizes the prognostic relevance of mRNA quantification in EpCAM-positive circulating tumor cells from early stage breast cancer patients. The authors conclude that detection of TWIST1 overexpression and stem-cell (CD24, CD44, ALDH1) transcripts in the EpCAM+ CTC fraction provides prognostic information in early stage breast cancer patients. Overexpression of TWIST1 in the EpCAM+ CTC fraction in the group of HR negative patients or with >3 positive lymph nodes is associated with worse prognosis.

Comments/Suggestions:

Authors should update the literature on role of CTC in early Breast Cancer and compare the findings in discussion (eg. SUCCESS Study).

We have now added in the revised manuscript in the introduction on page 2, lines 60-67 the following:" In early breast cancer …….. the detection rate of CTC [31]. However," and in discussion on page 12, lines 297-304 the following:" Relevant prognostic and predictive markers……… signalling pathways like EMT or CSC".   

Authors should mention about the selection criteria for candidate markers.

We selected TWIST1 as this is a very established EMT marker; for this reason we have developed already in 2011 an RT-qPCR assay for the absolute quantification of TWIST1-mRNA expression and we have validated this assay in EpCAM-positive cells isolated by early and metastatic breast cancer patients (Strati et.al BMC Cancer. 2011 Oct 4;11:422.). Concerning the selection of stem cell markers, this was based on publications by Al-Hajj M et.al. (Al-Hajj M, Wicha MS, Benito-Hernandez A, Morrison SJ, Clarke MF. Prospective identification of tumorigenic breast cancer cells. Proc Natl Acad Sci U S A. 2003 Apr 1;100(7):3983-8.) and Ginestier C et.al. (Ginestier C, Wicha MS. Mammary stem cell number as a determinate of breast cancer risk. Breast Cancer Res. 2007; 9(4):109), who have shown that the breast cancer stem cell phenotypes of: a) CD44+CD24−/low phenotype and b) the overexpression of aldehyde dehydrogenase 1 (ALDH1+) are able to form tumors in mice with high tumorigenic capacity.

We would like to thank the reviewer for this comment. We have now added in the revised manuscript in the Discussion on page 12 lines 295-305 the following:" we selected… tumorigenic capacity".

How was the cut-off for mRNA expression determined for each marker? Can authors deduce a composite score for independent (not correlated) markers?

As we already have mentioned in the section of materials and methods (lines 132-137) "all data were evaluated in respect to TWIST1, CD24, CD44, and ALDH1 expression by normalizing the EpCAM+ fraction of PBMCs to the expression of HPRT and the 2–ΔΔCt approach, as described in detail by Livak and Schmittgen [37]. A cut-off value was calculated as the mean of signals derived by samples of healthy individuals analysed in exactly the same way, plus 2SD for TWIST1, CD44, and ALDH1 transcripts and as the mean of signals derived by samples of healthy individuals minus 2SD for CD24." However, we cannot deduce a composite score for independent (not correlated) markers because firstly the data for TWIST1 overexpression and Stem Cell profile are obtained by two different and independent RT-qPCR assays and secondly because the levels of the TWIST1, CD24, CD44, and ALDH1 mRNA quantification varies in the EpCAM+ cells isolated from the same peripheral blood sample . These genes are expressed at different levels individually in healthy donors PBMCs, so we had to evaluate for each transcript mRNA over or under-expression based on individual cut-offs. 

Why and how the target genes were dichotomized as “low” and “high” expressors?

TWIST1, CD24, CD44, and ALDH1-mRNA is expressed both in EpCAM+ and EpCAM- cells. For this reason, target genes were dichotomized as “low” and “high” expressors, so as to discriminate the samples with high number of these gene transcripts from those with a low number of transcripts. As we have already mentioned in Materials and Methods (lines 132-137) "all data were evaluated in respect to TWIST1, CD24, CD44, and ALDH1 expression by normalizing the EpCAM+ fraction of PBMCs to the expression of HPRT and the 2–ΔΔCt approach, as described in detail by Livak and Schmittgen [37]. A cut-off value was calculated as the mean of healthy individual plus 2SD for TWIST1, CD44, and ALDH1 and as the mean of healthy individual minus 2SD for CD24." The mean of healthy individual plus 2SD refers to high expression and the mean of healthy individual minus 2SD refers to low expression of the target genes.

Authors may want to comment on limitations and future directions for the study.

We would like to thank the reviewer for this comment. We have now added the following paragraph in the revised manuscript on page 13, lines 337-341: "The main limitation of our study is that we examined the expression of only one EMT marker, TWIST1 in the EpCAM+ cells of early breast cancer patients. Since, there is a high heterogeneity in CTC, it may be possible that we have not detected CTC that express other mesenchymal markers like Vimentin or Snail.  We plan to extend this study by adding more gene expression markers in a biggest sample cohort and correlate our results to the clinical outcome."

What is sensitivity and specificity for extraction of EpCAM+ cells from 20ml of blood? What is RNA yield?

Recently we evaluated in detail the effect of pre-analytical conditions on gene expression analysis in CTCs by spiking 100 MCF-7 cells in 6 different commercially available blood collection tubes at different time points and under different storage conditions (Zavridou et.al. Clin Chem. 2018 Oct;64(10):1522-1533.). In this study we also performed spiking experiments (data not shown) using 10-100-1000 MCF7 in 10mL of PB and found that we can efficiently isolate 10 MCF-7 cells/10 mL of whole blood. The RNA extraction procedure that we always follow is described in detail in the section RNA extraction-cDNA synthesis (lines 103-111). The RNA yield of the EpCAM+ fraction ranged between 15-60 ng/μL.

Was control (HPRT) included with TWIST1 gene expression assay in the same reaction? How the mRNA expression was normalized?

RT-qPCR for the reference gene HPRT was not performed in the same reaction with the RT-qPCR for TWIST1. HPRT was included in a novel quadraplex RT-qPCR assay for the simultaneous quantification of CD24, CD44, ALDH1, and HPRT transcripts (reference gene) (line 113-114). TWIST1-mRNA expression was normalized based on the HPRT-mRNA expression that was run for the same samples on the same day. The accuracy and reproducibility of RT-qPCR assays for both TWIST1 and HPRT transcripts was evaluated by including in all runs the same cDNA as a positive control.

Minor Comments:

Authors may provide full form for the abbreviations used first time in text (eg. “HD” to Healthy Donor).

In the revised manuscript we have now added a list of abbreviations between lines 361-367.

Provide data values consistent across the manuscript (eg- Line 184- provide numbers corresponding to 25%)

In the revised manuscript we have now corrected this.

Provide detailed figure legend for 2A. Explain the heatmap (what the colors are standing for)?

We corrected the figure legend for 2A "Figure 2. A) Heat map of TWIST1, CD24, CD44 and ALDH1-mRNA quantification in the EpCAM+ CTC fraction from early stage breast cancer patients (n=100)". Red colour represents overexpression, while green colour indicates under-expression or lack of expression. Concerning the relapse or death red colour represents the relapse or death, while green colour indicates no relapse or alive status.

Provide complete figure legend for Figure 3.

We corrected the figure legend for Figure 3 as follows: "Figure 3. Kaplan–Meier estimates for early BrCa patients: A) DFI: TWIST1 overexpression, B) OS: TWIST1 overexpression, C) DFI: TWIST1 overexpression and number of affected lymph nodes, D) OS: TWIST1 overexpression and number of affected lymph nodes, E) DFI: TWIST1 overexpression and HR status, F) OS: TWIST1 overexpression and HR status."

Reviewer 3 Report

The manuscript by Strati et. colleagues addresses the interesting question of the role of CTCs expressing mesenchymal and/or stem cell traits.

The methods are sound, and importantly an adequate number of healthy donors has been considered to reliably define a cutoff value.

Whereas I understand how the cutoff values have been calculated, I have difficulties in understanding the data reported in Fig. 1. Assuming that the 2-DDct  method has been used , what was the reference sample for defining the fold changes reported in the figure. The HPRT gene should be the housekeeping gene used to normalize target gene expression both in the unknown sample as well as in the reference/calibrator sample. Also p values in Fig. 1b seem to be inverted

I would also have appreciated to see what is the added value of TWIST and stemness genes in addition to the classic CTC definition. No data are provided on CTC+ vs CTC negative samples.

Table 3 generates some confusion, since ER and PR variables are not statistically signifcant in univariate analysis and become significant when performing a multivariate analysis. Where the data checked for interactions?

Author Response

REVIEWER 3

Comments and Suggestions for Authors

The manuscript by Strati et. colleagues addresses the interesting question of the role of CTCs expressing mesenchymal and/or stem cell traits.

The methods are sound, and importantly an adequate number of healthy donors has been considered to reliably define a cutoff value.

Whereas I understand how the cutoff values have been calculated, I have difficulties in understanding the data reported in Fig. 1. Assuming that the 2-DDct method has been used, what was the reference sample for defining the fold changes reported in the Figure. The HPRT gene should be the housekeeping gene used to normalize target gene expression both in the unknown sample as well as in the reference/calibrator sample.

 We have not used a single reference sample for the normalization of our data, but for each gene we have used as a reference group the mean of all values derived from the healthy donor samples tested in exactly the same way as patient samples.

Also p values in Fig. 1b seem to be inverted.

The p values in Fig. 1b are not inverted, since CD24-/low contributes to the Stem cell profiles CD24-/low/CD44high and/or CD24-/low/ALDH1high

I would also have appreciated to see what is the added value of TWIST and stemness genes in addition to the classic CTC definition. No data are provided on CTC+ vs CTC negative samples.

Unfortunately we cannot answer this question, since we have not checked these samples by CellSearch, which is considered as the classic CTC definition. However in early breast cancer stages, to enumerate CTCs with the CellSearch would require a large volume of peripheral blood in addition to the 20 mL taken for this study. It was not our purpose to compare the detection of these markers with the number of CTCs. So we cannot evaluate our results in respect to CTC+ vs CTC negative samples.

Table 3 generates some confusion, since ER and PR variables are not statistically significant in univariate analysis and become significant when performing a multivariate analysis. Where the data checked for interactions?

According to Heinze et al, (Heinze G et.al. Five myths about variable selection. Transpl Int. 2017 Jan;30(1):6-10.), it is possible that non-significant variables on univariate analysis will become significant on multivariate analysis. More specifically, although it is true that regression coefficients are often larger in univariable models than in multivariable ones, also the opposite may occur, in particular if some variables (with all positive effects on the outcome) are negatively correlated. Moreover, univariable prefiltering, sometimes also referred to as “bivariable analysis,” does not add stability to the selection process as it is based on stochastic quantities, and can lead to overlooking important adjustment variables needed for control in an etiologic model. Although univariable prefiltering is traceable and easy to do with standard software, one should better completely forget about it as it is neither a prerequisite nor providing any benefits when building multivariable models